# Effect of the Freeze-Drying Process on the Physicochemical and Microbiological Properties of Mexican Kefir Grains

**Alicia Águeda Conde-Islas** [1] [iD]**, Maribel Jiménez-Fernández** [2]**, Denis Cantú-Lozano** [1] [iD]**,**
**Galo Rafael Urrea-García** [1] **and Guadalupe Luna-Solano** [1,*] [iD]

[1] Instituto Tecnológico de Orizaba, División de Estudios de Posgrado e Investigación, Oriente 9 No. 852 Orizaba, Veracruz 94320, Mexico; alicia_aconde@hotmail.com (A.Á.C.-I.); dencantu@gmail.com (D.C.-L.); galourrea@hotmail.com (G.R.U.-G.)

[2] Instituto de Ciencias Básicas, Universidad Veracruzana, Dr. Luis Castelazo s/n. Col. Industrial Animas Xalapa, Veracruz 91000, Mexico; maribeljimenezf@hotmail.com

[*] Correspondence: lupitals@hotmail.com; Tel.: +52-1-(272)-72-5-70-56

**Abstract:** The purpose of this study was to investigate how the properties of Mexican kefir grains (MKG) are affected by the operating parameters used in the freeze-drying process. The factors investigated were the freezing time (3–9 h), freezing temperature (−20 to −80 °C), pressure (0.2–0.8 mbar), and lyophilization time (5–20 h). The maximum range of change and one-way analysis of variance showed that lyophilization time and freezing time significant affects ($p < 0.05$) the response variables, residual moisture content and water activity, and pressure had a significant effect on the color difference and survival rate of probiotic microorganisms. The best drying conditions were a freezing time of 3 h, a freezing temperature of −20 °C, a pressure of 0.6 mbar, and a lyophilization time of 15 h. Under these conditions, we obtained a product with residual moisture content below 6%, water activity below 0.2, and survival rates above 8.5 log cfu per gram of lactic acid bacteria and above 8.6 log for yeast.

**Keywords:** freeze-drying; operating parameters; properties; kefir grains; probiotic

## 1. Introduction

Kefir grains are the culture used to produce kefir: a probiotic and popular beverage in the functional food market. This culture is a complex mixture of bacteria and yeasts that live in a symbiotic relationship [1–3]. These grains have an irregular form with a shape similar to that of a cauliflower, are yellow or white color, and have a variable size from 0.3 to 3.5 cm in diameter [4–6]. The grains are initially very small but increase in size during the fermentation process, but they can only grow from preexisting grains [7]. Their complex microbiological mixture depends on their origin, quality, and type of milk used for reproduction. As a result, obtaining a defined and uniform starter culture for the industrial kefir market is difficult [8]. For this reason, dehydration of kefir may provide a solution to increase the market value of this dairy product [9].

Probiotic foods are increasing in popularity in the marketplace [10] and they are defined as live organisms which, when administered in adequate amounts, provide health benefits to the host. To provide these benefits, a probiotic must contain at least 10 million colony-forming units per gram (cfu/g) [11]. This implies that the microorganisms must remain viable and maintain their characteristics from the beginning of the production process until they are consumed because probiotic microorganisms must survive throughout the production and storage processes [12,13].

Freeze-drying is a well-known dehydration process in the food and dairy industry where the major applications are the preservation of probiotics, starter cultures, and biological material. Compared to other drying techniques such as spray drying freeze-drying is an expensive process due to its relatively high investment, operation, and maintenance costs [14], but the freeze-drying process produces many benefits in the results obtained in terms of viability, quality, and storage time in dairy bacterial cultures [15–19]. Freeze-drying also has advantages in terms of the characteristics of the final product such as stability in its composition, intact nutrients, reduction of degradation in heat-sensitive products, reduction of chemical degradation, control of the final humidity, and reduction of water content at the end of the process to very low levels [20–22]. Freeze-drying is a process where water or another solvent is frozen, followed by its elimination from the sample in final steps of the lyophilization. This process combines in three steps: (1) the sample is frozen at low temperature, (2) primary drying occurs when the ice is sublimated under reduced pressure, and (3) secondary drying occurs as a desorption step where the residual moisture is removed or reduced to a low level. The objective of any kind of microbiological drying process is to enable cell survival and long storage [23–25]. Some studies [5,26–30] have examined freeze drying of kefir grains, analyzing the production of freeze-drying kefir cultures from whey and the effects of freeze-drying on the survival of microorganisms present in the grains with or without the use of cryoprotectants. Until now, no study has considered the effects of process variables on a stable and uniform starter culture with suitable characteristics in terms of viability, percent residual moisture content (%RMC), water activity ($a_w$), and color for the kefir market.

## 2. Materials and Methods

### 2.1. Kefir Grains Preparation

Fresh kefir grain biomass was obtained from local sources in Orizaba, Veracruz, Mexico. Propagation, activation, and adaptation to the medium were performed in a laboratory for 60 days. Ultrahigh temperature (UHT) milk samples were inoculated with kefir grains 10:90 (*w/v*) by 24-h cycles at a temperature of 26 °C. Prior to freeze-drying, kefir grains were selected and washed with distilled water until all residues of milk were removed [31]. Kefir grain biomass viability was important for experimental data validity [32].

### 2.2. Experimental Design and Freeze-Drying

Freeze-drying of Mexican kefir grains (MKG) was carried out in a freeze dryer (Labconco Equipment Co., KS, USA) with capacity of 12 L and an ultralow freezer (Environmental Equipment, Cincinnati, OH, USA). The selected levels for each affecting factor are shown in Table 1. The freezing time was set to 3, 5, 7, or 9 h; freezing temperature was set to −20, −40, −60, or −80 °C; and pressure was 0.2, 0.4, 0.6, or 0.8 mbar. For this study 5, 10, 15, and 20 h were selected as the lyophilization times. The $L_{16}\ 4^4$ orthogonal design used is shown in Table 2.

**Table 1.** Factors and levels used in the freeze-drying experiments.

| Level | Factor | | | |
|---|---|---|---|---|
| | *Ft* (h) | *Fte* (°C) | *P* (mbar) | *Lt* (h) |
| 1 ($K_1$) | 3 | −20 | 0.2 | 5 |
| 2 ($K_2$) | 5 | −40 | 0.4 | 10 |
| 3 ($K_3$) | 7 | −60 | 0.6 | 15 |
| 4 ($K_4$) | 9 | −80 | 0.8 | 20 |

*Ft*: freezing time; *Fte*: freezing temperature; *P*: pressure; *Lt*: lyophilization time.

**Table 2.** $L_{16}4^4$ orthogonal experimental design sheet.

| Experiment | Factor | | | |
|---|---|---|---|---|
| | *Ft* (h) | *Fte* (°C) | *P* (mbar) | *Lt* (h) |
| 1 | 1 | 1 | 1 | 1 |
| 2 | 1 | 2 | 2 | 2 |
| 3 | 1 | 3 | 3 | 3 |
| 4 | 1 | 4 | 4 | 4 |
| 5 | 2 | 1 | 2 | 3 |
| 6 | 2 | 2 | 1 | 4 |
| 7 | 2 | 3 | 4 | 1 |
| 8 | 2 | 4 | 3 | 2 |
| 9 | 3 | 1 | 3 | 4 |
| 10 | 3 | 2 | 4 | 3 |
| 11 | 3 | 3 | 1 | 2 |
| 12 | 3 | 4 | 2 | 1 |
| 13 | 4 | 1 | 4 | 2 |
| 14 | 4 | 2 | 3 | 1 |
| 15 | 4 | 3 | 2 | 4 |
| 16 | 4 | 4 | 1 | 3 |

*2.3. Analytical Methods*

The residual moisture content (% RMC) of dried kefir grains was measured using the MA35 moisture analyzer (Sartorius AG, Göttingen, Germany). Samples of 0.5 g were heated to 65 °C until constant weight. Water activity ($a_w$) was determined at 25 °C using an Aqualab water activity meter (series 3, Decagon Devices Inc., Pullman, WA, USA). Color measurements were recorded using a Hunter Lab MiniScan XE Plus (Hunter Associates Laboratory Inc., Reston, VA, USA) and results are expressed in Hunter Lab Units *L*\* (lightness/darkness), *a*\* (redness/greenness), and *b*\* (yellowness/blueness). Survival rate of lactic acid bacteria (LAB) and yeasts before and after lyophilization are expressed as log (cfu/g), which was determined using the pour plate method. Decimal dilutions with peptone water were prepared from the kefir grains and LAB were quantified on Man Rogosa and Sharpe agar (MRS) (Dibico, Cuautitlan Izcalli, Mexico) at 35 °C for 3 days. Yeasts were evaluated on yeast peptone dextrose medium (YPD) (Dibico, Cuautitlan Izcalli, Mexico) after incubation at 25 °C for 5 days.

*2.4. Statistical Analysis*

Significance of the affecting factor was determined by analysis of variance (ANOVA) at $p < 0.1$ and $F > F$crit (critical factor) = 2.605525, and at $p < 0.05$ and $F > F$crit = 3.490295. The importance of the affecting factor in range analysis was determined by the maximum changing range of dried final product properties [33]. Multiple regression was used to predict the behavior between the final properties of the dried product and the affecting factors, i.e., the final properties as a dependent variable and four parameters—freezing time (*Ft*), freezing temperature (*Fte*), pressure (*P*), and lyophilization time (*Lt*)—as independent variables in the analysis.

**3. Results and Discussion**

*3.1. Residual Moisture Content of Dried Mexican Kefir Grains*

The residual moisture content of dried MKG is an important factor that affects the long-term stability of the product and its shelf life. Our analysis of the % RMC of the dried MKG is shown in Table 3. The results show that freezing time, freezing temperature, and pressure in the four selected levels did not have a significant effect on this response variable, but lyophilization time did have a significant effect for all four levels ($p = 0.00163 < 0.1$ and $F = 9.62136 > F$crit = 2.605525; $p = 0.00163 < 0.05$, and $F = 9.62136 > F$crit 3.490295). This finding is consistent with the conclusions of other researchers [34–38], who indicated that secondary drying (desorption) in the freeze-drying process plays a significant role in the desired moisture content of the final product.

**Table 3.** Data analysis of dried Mexican kefir grain (MKG) residual moisture content (% RMC).

| Data Analysis % RMC | Factor | | | |
|---|---|---|---|---|
| | *Ft* (h) | *Fte* (°C) | *P* (mbar) | *Lt* (h) |
| $K_1$ | 23.01 | 30.55 | 21.46 | 188.69 |
| $K_2$ | 99.40 | 62.26 | 77.36 | 46.07 |
| $K_3$ | 70.77 | 79.11 | 83.95 | 15.66 |
| $K_4$ | 73.06 | 94.32 | 83.46 | 15.82 |
| $R'$ | 76.39 | 63.78 | 62.49 | 173.03 |
| $F$ | 0.46713 | 0.33514 | 0.41678 | 9.62136 |
| *p* Value | 0.71063 | 0.80025 | 0.74417 | 0.00163 |

MKG: Mexican kefir grains; *Ft*: freezing time; *Fte*: freezing temperature; *P*: pressure; *Lt*: lyophilization time; $K_1$: level 1; $K_2$: level 2; $K_3$: level 3; $K_4$: level 4; $R'$: changing range.

The max changing range (173.03) of the lyophilization time in this analysis was the largest among the range of the four variables ($R'$). This indicates that the lyophilization time plays an important role in affecting the % RMC of dried MKG. The range (76.39) for freezing time was larger than those of the remaining two factors. This shows that freezing time is the second-most important factor that affects the residual moisture in the dried product.

According to the multiple regression analysis of the $L_{16}4^4$ orthogonal experiments results, the regression equation for the residual moisture content of dried MKG is obtained as follows.

$$\text{RMC (\%)} = 12.3 + 12.7X_1 - 10.59X_2 + 36.60X_3 - 38.85X_4 - 2.74X_1^2 + 4.29X_1X_2 - 0.97X_1X_3 \\ - 2.460X_1X_4 + 1.143X_2^2 - 0.80X_2X_4 - 3.12X_3^2 - 4.348X_3X_4 + 8.924X_4^2 \tag{1}$$

In this equation, the square of the correlation coefficient ($R^2$) was 0.9974. The coefficient (38.85) of lyophilization time was the largest among the coefficients of the four variables. This also indicates that the lyophilization time plays an important role in affecting the % RMC of dried MKG. On the other hand, the coefficients 36.60 and 12.70 for pressure and freezing time, respectively, were the largest than those of the remaining factors it can be showed with *p*-values for each coefficient.

According to Equation (1), the predicted % RMC values of dried MKG with different freezing times and pressures are shown in Figure 1. The lyophilization time and freezing temperature in the equation were 5 h and −20 °C, respectively. Figure 1 shows that % RMC increases more with increased pressure and with a freezing time of 5 h. However, we observed that as when the pressure was less than 0.1, the % RMC decreased. Table 4 shows the experimental and predicted % RMC of dried MKG and the error differences. The smallest value obtained in the experimental results was 2.38 with a freezing time of 5 h, freezing temperature of −40 °C, pressure of 0.2 mbar, and lyophilization time of 20 h. The error relative difference between the experimental and predicted % RMC was −15.29%.

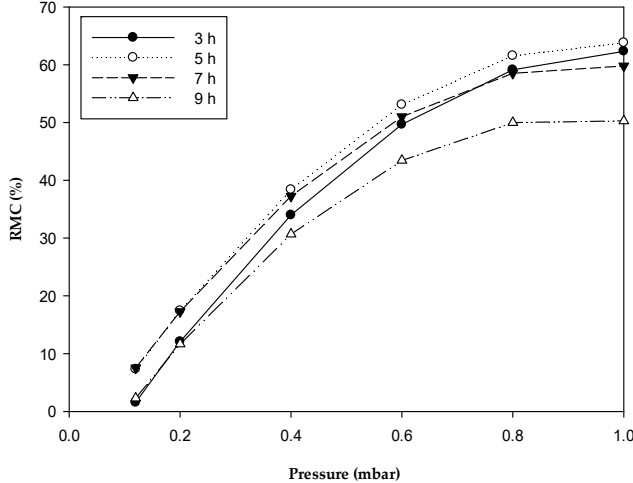

**Figure 1.** Predicted residual moisture content of dried MKG at different freezing time and pressure.

Table 4. Experimental (Exp-R) and predicted results (Pre-R) of dried MKG properties.

| Experiment | Residual Moisture Content(% RMC) | | | Water Activity(a_w) | | | Color Difference | | | | | | Log (cfu/g) of LAB | | | Log (cfu /g) of Yeasts | | |
|---|---|---|---|---|---|---|---|---|---|---|---|---|---|---|---|---|---|---|
| | | | | | | | Exp-R | | | | | | | | | | | |
| | Exp-R | Pre-R | Error (%) | Exp-R | Pre-R | Error (%) | $L^*$ | $a^*$ | $b^*$ | $\Delta E$ | Pre-R | Error (%) | Exp-R | Pre-R | Error (%) | Exp-R | Pre-R | Error (%) |
| 1 | 11.84 | 12.08 | −2.02 | 0.173 | 0.177 | −2.20 | 44.02 | −1.13 | 6.40 | 15.5 | 14.77 | 4.73 | 4.99 | 4.72 | 5.37 | 5.22 | 4.64 | 11.24 |
| 2 | 4.84 | 5.50 | −13.55 | 0.111 | 0.121 | −9.19 | 38.42 | −0.98 | 5.58 | 6.23 | 7.66 | −22.92 | 8.03 | 8.27 | −2.94 | 5.63 | 6.94 | −23.24 |
| 3 | 3.08 | 2.51 | 18.47 | 0.114 | 0.105 | 8.07 | 39.96 | −1.12 | 6.43 | 7.92 | 6.5 | 17.89 | 8.56 | 8.32 | 2.85 | 8.67 | 7.44 | 14.24 |
| 4 | 3.25 | 3.12 | 3.88 | 0.131 | 0.128 | 2.60 | 50.58 | −1.44 | 7.49 | 10.55 | 11.3 | −7.13 | 4.61 | 4.87 | −5.68 | 5.44 | 6.11 | −12.42 |
| 5 | 4.84 | 3.23 | 33.24 | 0.111 | 0.089 | 20.00 | 36.73 | −1.34 | 4.85 | 3.93 | 4.86 | −23.59 | 6.81 | 7.45 | −9.43 | 8.44 | 9.10 | −7.81 |
| 6 | 2.38 | 2.74 | −15.29 | 0.137 | 0.142 | −3.65 | 42.30 | −1.45 | 6.79 | 10.98 | 10.82 | 1.48 | 8.20 | 8.06 | 1.76 | 8.27 | 8.19 | 1.03 |
| 7 | 65.33 | 65.08 | 0.38 | 0.983 | 0.98 | 0.35 | 39.11 | −1.43 | 5.76 | 7.92 | 8.05 | −1.68 | 8.35 | 8.47 | −1.49 | 8.4 | 8.55 | −1.71 |
| 8 | 26.85 | 28.56 | −6.35 | 0.494 | 0.518 | −4.86 | 42.60 | −1.30 | 5.71 | 8.95 | 8.00 | 10.59 | 8.42 | 7.76 | 7.86 | 8.68 | 8.08 | 6.94 |
| 9 | 2.94 | 4.64 | −57.86 | 0.136 | 0.161 | −18.31 | 38.65 | −0.91 | 5.53 | 12.74 | 11.77 | 7.61 | 8.64 | 7.99 | 7.56 | 8.67 | 8.03 | 7.36 |
| 10 | 3.95 | 4.36 | −10.43 | 0.210 | 0.216 | −2.95 | 35.46 | −1.08 | 4.80 | 7.13 | 6.89 | 3.37 | 6.49 | 6.31 | 2.85 | 6.20 | 6.08 | 1.92 |
| 11 | 3.45 | 3.18 | 7.91 | 0.092 | 0.088 | 4.02 | 44.60 | −1.11 | 6.61 | 22.17 | 22.36 | −0.83 | 6.59 | 6.75 | −2.35 | 7.43 | 7.59 | −2.04 |
| 12 | 60.43 | 58.85 | 2.62 | 0.986 | 0.964 | 2.27 | 37.98 | −1.51 | 5.57 | 9.34 | 10.25 | −9.70 | 8.48 | 9.10 | −7.35 | 8.48 | 9.14 | −7.77 |
| 13 | 10.93 | 9.86 | 9.74 | 0.557 | 0.543 | 2.53 | 49.33 | −1.52 | 7.93 | 15.60 | 16.17 | −3.62 | 7.13 | 7.59 | −6.47 | 4.29 | 4.71 | −9.82 |
| 14 | 51.09 | 52.64 | −3.04 | 0.972 | 0.995 | −2.37 | 41.17 | −1.17 | 6.78 | 8.80 | 7.89 | 10.3 | 7.54 | 6.91 | 8.34 | 8.14 | 7.54 | 7.42 |
| 15 | 7.25 | 5.86 | 19.21 | 0.107 | 0.087 | 18.60 | 42.62 | −1.42 | 5.84 | 3.80 | 4.65 | −22.34 | 5.04 | 5.63 | −11.61 | 5.95 | 6.55 | −10.15 |
| 16 | 3.79 | 5.03 | −32.72 | 0.076 | 0.093 | −22.63 | 38.92 | −1.09 | 5.16 | 8.27 | 7.63 | 7.74 | 7.90 | 7.40 | 6.37 | 8.02 | 7.60 | 5.22 |

## 3.2. Water Activity of Dried Mexican Kefir Grains

Water activity ($a_w$) in the dried MKG is one the most important factors that affect the quality and probiotic viability during the storage of the dried product because, during the drying processes, the decrease in water activity damages the bacterial structures, decreasing their viability [39]. If the $a_w$ is below 0.60, complete inhibition of microbial growth and lipid oxidation are assured in food products during storage [40]. The water activity is provided in Table 4. For most of the experiments, the water activity was below 0.6.

The data obtained in this study are analyzed in Table 5. The water activity of the dried MKG was significantly influenced ($p < 0.1$ and $F = 6.64924 > F$crit $= 2.605525$ and $p = 0.00163 < 0.05$ and $F = 6.64924 > F$crit $3.490295$) by the four levels of the lyophilization time. This finding is consistent with previous research showing that during freeze-drying, extremely low water activity values are obtained because free water is frozen and low drying temperatures prevent thermal and enzymatic degradation of the final product [41,42].

**Table 5.** Data analysis of dried Mexican kefir grains (MKG) water activity ($a_w$).

| Data Analysis for $a_w$ | Factor | | | |
| --- | --- | --- | --- | --- |
| | *Ft* (h) | *Fte* (°C) | *P* (mbar) | *Lt* (h) |
| $K_1$ | 0.529 | 0.977 | 0.478 | 3.114 |
| $K_2$ | 1.725 | 1.430 | 1.315 | 1.254 |
| $K_3$ | 1.424 | 1.295 | 1.716 | 0.511 |
| $K_4$ | 1.712 | 1.687 | 1.880 | 0.511 |
| $R'$ | 1.196 | 0.711 | 1.402 | 2.603 |
| $F$ | 0.60391 | 0.15021 | 0.77428 | 6.64924 |
| *p* Value | 0.62489 | 0.92756 | 0.53045 | 0.00678 |

The maximum rate of change (2.603) for the lyophilization time in this analysis was the largest among the range of the four variables. This indicates that the lyophilization time plays an important role in affecting the $a_w$ of dried MKG. The range (1.402) obtained for pressure was larger than those of the remaining two factors shown, which means that pressure is another factor that significantly affects this response variable.

The equation for the water activity using multiple regression is

$$\begin{aligned} a_w = {} & 0.246 + 0.033X_1 - 0.098X_2 + 0.319X_3 - 0.3560X_4 - 0.0074X_1^2 + 0.0405X_1X_2 + 0.0441X_1X_3 \\ & - 0.0640X_1X_4 + 0.0307X_2^2 - 0.0321X_2X_4 - 0.0260X_3^2 - 0.0692X_3X_4 + 0.1162X_4^2 \end{aligned} \quad (2)$$

The $R^2$ for Equation (2) was 0.9979. The coefficient of the lyophilization time (0.3560) indicates that the lyophilization time plays the most important role in affecting the water activity of the dried MKG. The relationship between freezing time and pressure was obtained using Equation (2) when the freezing temperature and lyophilization time were −20 °C and 5 h, respectively (Figure 2). We found that the water activity decreased to almost zero under low pressure and a freezing time of 3 h when the other two factors were kept constant.

In Table 4, the experimental water activity is compared with the predicted water activity. The smallest value obtained in the experimental result was 0.076 with a freezing time of 9 h, freezing temperature of −80 °C, pressure of 0.2 mbar, and lyophilization time of 15 h. The error relative difference between the experimental and predicted $a_w$ was −22.63% in test number 16.

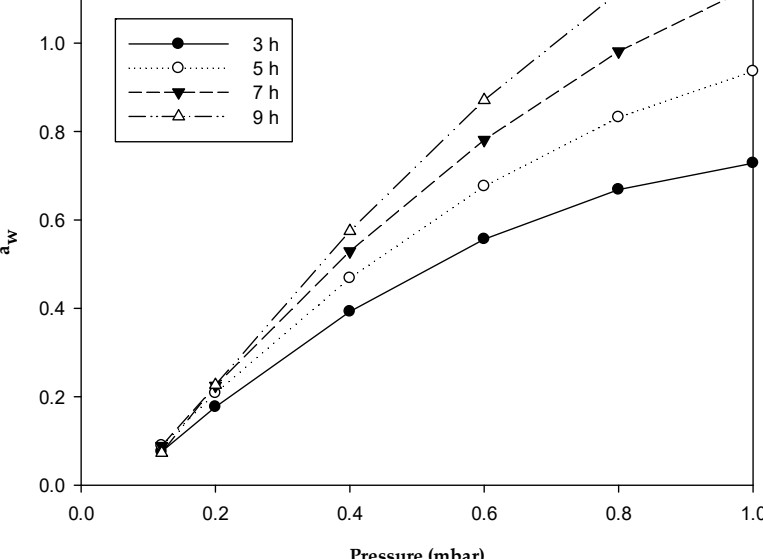

**Figure 2.** Predicted water activity of dried MKG at different freezing times and pressures.

### 3.3. Color Difference of Dried Mexican Kefir Grains

Color is the first judgment and major quality attribute in dried foods, because its change indicates that certain components in the product might have deteriorated [43–45]. In this study, the reference standard was a fresh sample of MKG, for which the *L**, *a**, and *b** values were 35.28, −1.15, and 2.84, respectively. The analysis of color data is shown in Table 6. In this table, *R'* is the maximum difference among the four levels for each factor. The larger the *R'*, the more significant the effects. Table 6 shows that lyophilization time affects the lightness (*L** value), whereas *a** is mostly affected by pressure. However, yellowness (*b** value) is more influenced by lyophilization time and pressure than the other two variables: freezing temperature and lyophilization time. These results differ from those reported in other studies that reported that color parameters are affected by freezing and drying time [46].

The results in Table 6 reveal that *L**, *a**, *b**, and ΔE (color difference) of dried MKG are affected by pressure. This can be noted, for example, in the analysis of the color difference influenced by pressure ($p = 0.07176 < 0.1$ and F = 3.01767 > Fcrit = 2.605525), whereas the other three parameters were not significant ($p < 0.1$ and $F > Fcrit = 2.605525$ and $p < 0.05$ and $F > Fcrit = 3.490295$) for this response variable. This change is seen in *R'*.

The equation for color difference (ΔE) of the dried product using the multiple regression analysis is

$$\Delta E = 16.2 + 11.02X_1 + 18.15X_2 - 21.51X_3\ -7.27X_4 + 0.61X_1^2 - 5.77X_1X_2 - 1.23X_1X_3$$
$$+ 1.590X_1X_4\ -0.257X_2^2 - 0.07X_2X_3 - 1.45X_2X_4 + 3.0X_3^2 + 1.754X_3X_4 \tag{3}$$

The $R^2$ for Equation (3) is 0.9666. Figure 3 provides the relationship among color difference (ΔE), freezing time, and pressure at freezing temperature −20 °C and lyophilization time of 5 h. At the same freezing time, the predicted color difference decreases with pressure increase until a pressure of 0.6 mbar, and remains constant up to a pressure of 0.8. The minimum ΔE is obtained with freezing time from 3 h. The coefficient of pressure, 21.51, indicates that the pressure plays the most important role in affecting color difference of the dried MKG.

**Table 6.** Data analysis of MKG color difference (ΔE).

| Data Analysis for ΔE | | Factor | | | |
|---|---|---|---|---|---|
| | | *Ft* (h) | *Fte* (°C) | *P* (mbar) | *Lt* (h) |
| *L*\* | $K_{L1}$ | 172.98 | 168.72 | 169.84 | 162.28 |
| | $K_{L2}$ | 160.74 | 157.35 | 155.75 | 174.95 |
| | $K_{L3}$ | 156.69 | 166.28 | 162.37 | 151.07 |
| | $K_{L4}$ | 172.04 | 170.08 | 174.47 | 174.15 |
| | $R'_L$ | 16.29 | 12.73 | 18.72 | 23.08 |
| | F | 0.91679 | 0.40606 | 0.94622 | 2.23689 |
| | *p* Value | 0.46201 | 0.75143 | 0.44902 | 0.13646 |
| *a*\* | $K_{a1}$ | −4.67 | −4.90 | −4.78 | −5.24 |
| | $K_{a2}$ | −5.52 | −4.68 | −5.25 | −4.91 |
| | $K_{a3}$ | −4.61 | −5.08 | −4.50 | −4.63 |
| | $K_{a4}$ | −5.20 | −5.34 | −5.47 | −5.22 |
| | $R'_a$ | 0.91 | 0.66 | 0.97 | 0.61 |
| | F | 1.28314 | 0.44649 | 1.30616 | 0.48155 |
| | *p* Value | 0.32468 | 0.72427 | 0.31764 | 0.70120 |
| *b*\* | $K_{b1}$ | 25.90 | 24.71 | 24.96 | 24.51 |
| | $K_{b2}$ | 23.11 | 23.95 | 21.84 | 25.83 |
| | $K_{b3}$ | 22.51 | 24.64 | 24.45 | 21.24 |
| | $K_{b4}$ | 25.71 | 23.93 | 25.97 | 25.65 |
| | $R'_b$ | 3.39 | 0.78 | 4.13 | 4.59 |
| | F | 0.94204 | 0.04580 | 0.96255 | 1.57480 |
| | *p* Value | 0.45084 | 0.98634 | 0.44197 | 0.24680 |
| ΔE | $K_{\Delta E1}$ | 40.20 | 47.77 | 56.92 | 41.56 |
| | $K_{\Delta E2}$ | 31.78 | 33.14 | 23.30 | 52.95 |
| | $K_{\Delta E3}$ | 51.38 | 41.80 | 38.41 | 27.25 |
| | $K_{\Delta E4}$ | 36.47 | 37.11 | 41.20 | 38.07 |
| | $R'_{\Delta E}$ | 19.61 | 14.63 | 33.62 | 25.71 |
| | F | 0.75283 | 0.39523 | 3.01767 | 1.36381 |
| | *p* Value | 0.54157 | 0.75880 | 0.07176 | 0.30074 |

MKG: Mexican kefir grains; $K_{L1}$: lightness level 1; $K_{L2}$: lightness level 2; $K_{L3}$: lightness level 3; $K_{L4}$: lightness level 4; $R'_L$: lightness changing range; $K_{a1}$: redness level 1; $K_{a2}$: redness level 2; $K_{a3}$: redness level 3; $K_{a4}$: redness level 4; $R'_a$: redness changing range; $K_{b1}$: yellowness level 1; $K_{b2}$: yellowness level 2; $K_{b3}$: yellowness level 3; $K_{b4}$: yellowness level 4; $R'_b$: yellowness changing range; $K_{\Delta E1}$: color difference level 1; $K_{\Delta E2}$: color difference level 2; $K_{\Delta E3}$: color difference level 3; $K_{\Delta E4}$: color difference level 4; $R'_{\Delta E}$: color difference changing range.

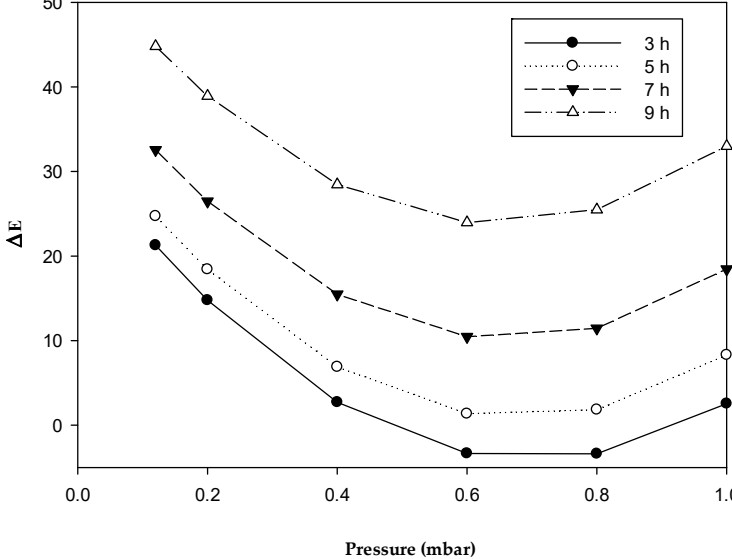

**Figure 3.** Predicted color difference (ΔE) of dried MKG at different freezing times and pressures.

Table 4 compares the experimental ΔE of dried MKG with the predicted ΔE. The lowest ΔE was 3.80 (Experiment 15) at freezing time 9 h, freezing temperature −60 °C, pressure 0.4 mbar, and lyophilization time 20 h. The maximum ΔE was 22.17 (Experiment 11). Figure 4 shows visual color images of MKG; the color of dried MKG with the lowest ΔE (Experiment 15) was very close to that of the fresh sample.

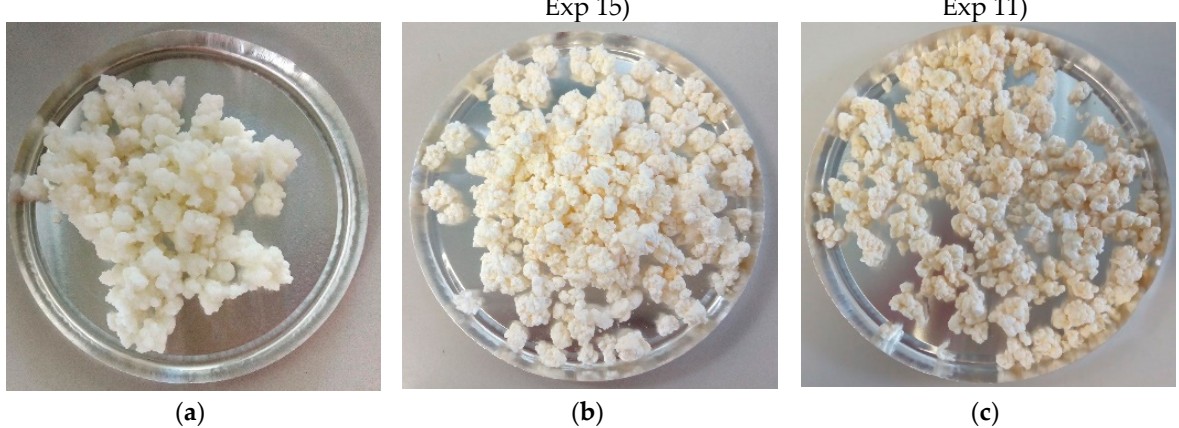

(**a**)      (**b**)      (**c**)

**Figure 4.** Visual color images of MKG: (**a**) fresh and dried MKG in (**b**) Experiment 15 (least change in color) and (**c**) Experiment 11 (most change in color).

### 3.4. Survival of Probiotic Microorganism of Dried Mexican Kefir Grains

The survival rate of microorganisms after freeze-drying requires adequate stabilization processes to avoid bacterial damage or death [47]. This is the most important requirement for a product to be considered probiotic. However, the survival rate of microorganisms easily decreases due to the stress to which bacteria are subjected during freeze-drying. For a food to be considered probiotic, it must contain at least $10^6$ colony forming units per gram (6 log cfu/g) of food. Counts above 8 log (cfu/g) have been suggested to ensure probiotic effects [48–50]. In this research, the survival rate of microorganisms was investigated under different experimental drying conditions. In this analysis, the initial value of survival of probiotic microorganisms in the fresh sample of MKG was 9.05 and log (cfu/g) for LAB and 8.90 log (cfu/g) for yeasts.

The results in Table 7 show that $R'$ is the maximum difference among the four levels for each factor. The max changing range (6.57) of pressure was the largest among the range of the four variables. This indicates that pressure plays an important role in affecting the survival LAB of dried MKG. The range (5.60) for freezing time was larger than those of the remaining two factors. This shows that freezing time is another important factor that affects the survival LAB in the dried product.

**Table 7.** Data analysis of dried MKG log (cfu/g) of lactic acid bacteria (LAB).

| Data Analysis for Log (cfu/g) of LAB | Factors | | | |
|---|---|---|---|---|
| | *Ft* (h) | *Fte* (°C) | *P* (mbar) | *Lt* (h) |
| $K_1$ | 26.19 | 27.57 | 27.67 | 29.37 |
| $K_2$ | 31.79 | 30.27 | 28.36 | 30.17 |
| $K_3$ | 30.20 | 28.54 | 33.16 | 29.76 |
| $K_4$ | 27.61 | 29.40 | 26.59 | 26.48 |
| $R'$ | 5.60 | 2.69 | 6.57 | 3.69 |
| $F$ | 0.81998 | 0.14854 | 1.16898 | 0.32509 |
| *p* Value | 0.50748 | 0.92864 | 0.36213 | 0.80726 |

Table 7 shows that although the four factors do not have a significant influence ($p < 0.1$ and $F < F$crit = 2.605525, neither $p > 0.05$ nor F < Fcrit = 3.490295) on the survival log (cfu/g) of yeasts of

dried MKG, the greatest effect was observed in $R'$, as this variable was affected by the pressure and freezing time.

The equation for log (cfu/g) LAB of the dried MKG using multiple regression analysis is

$$\begin{aligned}
\text{Log (cfu/g) of LAB} = {} & 0.06 - 2.88X_1 + 3.09X_2 + 2.332X_3 + 3.52X_4 + 0.504X_1^2 - 0.157X_1X_4 \\
& + 0.103X_2^2 - 0.510X_2X_3 - 0.907907X_2X_4 - 0.433X_3X_4
\end{aligned} \quad (4)$$

In this equation, $R^2$ is 0.8851. The relationship between the log (cfu/g) LAB and pressure was obtained using Equation (4) when freezing temperature and lyophilization time were maintained at $-20\,^{\circ}\text{C}$ and 5 h, respectively. The predicted log (cfu/g) of LAB is shown in Figure 5. The log (cfu/g) of LAB increases with increasing pressure increases and decreasing freezing time. The maximum value was obtained for a pressure of 1 mbar and 3 h freezing time. With higher pressures, the survival of LAB increases more than two logarithmic cycles.

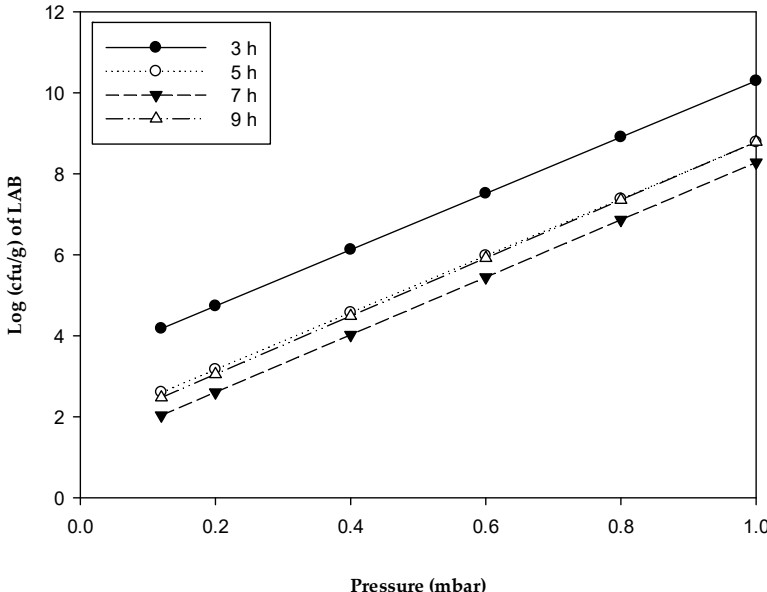

**Figure 5.** Predicted log (cfu/g) of LAB of dried MKG at different freezing times and pressures.

The experimental and predicted log (cfu/g) of LAB values of dried MKG are shown in Table 4. The highest value was 8.64 log (cfu/g) of LAB, less than 0.5 cycle logarithmic of the fresh sample before drying (9.05 log (cfu/g) of LAB). This was obtained at a freezing time 7 h, freezing temperature of $-20\,^{\circ}\text{C}$, pressure of 0.6 mbar, and lyophilization time of 20 h. The relative error difference between the experimental and predicted log (cfu/g) of LAB was 7.56%. Freeze-drying had a negative effect on the survival rate of 4.53%. This result is different from other researchers who reported a negative effect of 7% on the viability of LAB without the use of cryoprotectants during the freeze-drying process [51,52].

From the results presented in Table 8 for yeasts show that $R'$ is the maximum difference among the four levels for each factor. The max changing range (9.83) of pressure was the largest among the ranges of the four variables. This indicates that pressure plays an important role in yeast survival in dried MKG. The range of 8.83 for freezing time was larger than those of the remaining two factors. This shows that freezing time is another important factor affecting the survival of the yeasts in the dried product.

**Table 8.** Data analysis of dried MKG log (cfu/g) of yeasts.

| Data Analysis for Log (cfu/g) of Yeasts | Factor | | | |
|---|---|---|---|---|
| | *Ft* (h) | *Fte* (°C) | *P* (mbar) | *Lt* (h) |
| $K_1$ | 24.97 | 26.63 | 28.94 | 25.02 |
| $K_2$ | 33.79 | 28.25 | 28.51 | 21.75 |
| $K_3$ | 30.78 | 30.46 | 34.16 | 31.48 |
| $K_4$ | 26.40 | 30.61 | 24.33 | 28.33 |
| $R'$ | 8.83 | 3.98 | 9.83 | 6.58 |
| *F* | 2.23175 | 0.34546 | 2.19270 | 0.53571 |
| *p* Value | 0.13706 | 0.79307 | 0.14177 | 0.66659 |

Table 8 shows that although the four factors do not have a significant influence ($p > 0.1$ and $F < F$crit = 2.605525, neither $p > 0.05$ nor $F < F$crit = 3.490295) on the survival log (cfu/g) of yeasts of dried MKG, the greatest effect can be observed in $R'$, where this variable is affected by the pressure and freezing time.

Multiple regression analysis for log (cfu/g) of yeasts is as follows

$$\text{Log (cfu/g) of yeasts} = 2.52 + 3.35X_1 - 0.71X_2 + 3.90X_3 + 2.57X_4 - 0.32X_1^2 + 2.77X_1X_2 - 0.656X_1X_3 - 0.348X_1X_4 + 0.080X_2^2 + 0.150X_2X_3 - 0.42X_2X_4 - 0.375X_3^2 - 0.343X_3X_4 \tag{5}$$

In this equation, the $R^2$ was 0.8017. The log (cfu/g) of yeast determined using Equation (5) is shown in Figure 6. The freezing temperature was set to −20 °C and lyophilization time to 5 h. Figure 6 shows that the maximum predicted survival of the yeast is reached when the pressure increases to 0.8 mbar and freezing temperature is 3 h. With higher values of pressure and a higher freezing time of 3 h, the survival of the yeast decreases more than two logarithmic cycles.

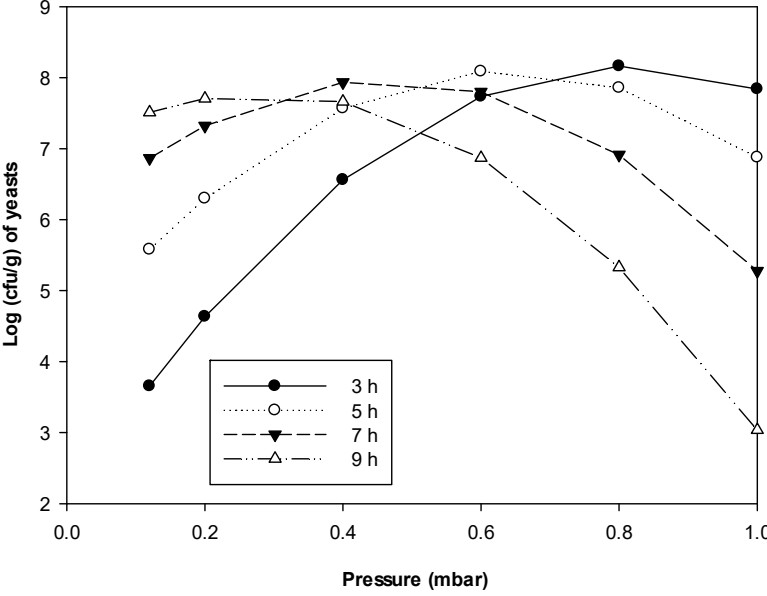

**Figure 6.** Predicted log (cfu/g) of yeasts of dried MKG at different freezing times and pressures.

The experimental and predicted log (cfu/g) values of yeasts of dried MKG are shown in Table 4. The highest value of 8.67 log (cfu/g) for yeasts is less than 0.5 cycle logarithmic of the fresh sample before drying (8.90 log (cfu/g) of yeasts). This was obtained at freezing times of 3 to 7 h, freezing temperatures of −20 to −60 °C, pressure of 0.6 mbar, and lyophilization time of 15 to 20 h. The relative error differences between the experimental and predicted log (cfu/g) of yeasts were 14.24 and 7.36% for tests No. 3 and 9, respectively. Freeze-drying had a negative effect on the survival rate of 2.60%;

this result is similar to that reported in other studies, which also reported a negative effect, even with the use of cryoprotectants [53,54].

### 3.5. Optimization of the Freeze-Drying Parameters for Dried Mexican Kefir Grains

Because the residual moisture, water activity, and survival of probiotic bacteria are the most important factors for dried MKG, optimization of the freeze-drying considers these three factors, with the goal of obtained values for % RMC below 6, $a_w$ below 0.60, and at least $10^6$ colony forming units per gram (6 log cfu/g) of probiotic microorganism. The optimum process for desirable moisture, water activity, and survival of probiotic microorganisms is shown in Table 9. Optimal levels for those parameters are the same: freezing time of 3 h, freezing temperature of −20 °C, pressure 0.6 mbar, and lyophilization time of 15 h. The residual moisture in the dried product needs to be below 6% [55]. In this study, a residual moisture content in dried MKG of less than 6% was obtained. As shown in Table 4, the pressure of 0.6 mbar was chosen as optimal to prevent probiotic bacterial loss. At a pressure of 0.2 mbar, we observed a lower survival in LAB and yeasts, with the lowest value being 4.99 log (cfu/g) for LAB and 5.22 log (cfu/g) for yeasts. Therefore, the pressure of 0.6 mbar was selected as optimal. Under these optimal conditions, the properties of dried MKG were: residual moisture 5.03%, $a_w$ 0.167, ΔE 7.79, LAB 8.5 log (cfu/g), and yeasts 8.6 log (cfu/g).

**Table 9.** Optimal factors for each properties of dried MKG.

| Property of Dried MKG | Factor | | | |
|---|---|---|---|---|
| | *Ft* (h) | *Fte* (°C) | *P* (mbar) | *Lt* (h) |
| RMC | Level 1 | Level 1 | Level 3 | Level 3 |
| $a_w$ | Level 1 | Level 1 | Level 3 | Level 3 |
| Log (cfu/g) of LAB | Level 1 | Level 1 | Level 3 | Level 3 |
| Log (cfu/g) of yeasts | Level 1 | Level 1 | Level 3 | Level 3 |

### 4. Conclusions

We freeze-dried Mexican kefir grains in a compact freeze-dryer. The $L_{16}$ orthogonal experimental design was applied to find the optimal operating conditions for obtaining a product suitable for the market. The residual moisture and water activity were mainly influenced by the operating lyophilization time ($p < 0.05$). These response variables decreased when the lyophilization time decreased. The color difference and survival rate of probiotic bacteria were significantly affected by pressure. The optimal conditions for dried MKG freeze-drying are a freezing time of 3 h, a freezing temperature of −20 °C, a pressure of 0.6 mbar, and a lyophilization time of 15 h. Under these optimal conditions, the properties of dried MKG were a residual moisture of 5.03%, an $a_w$ of 0.167, an ΔE of 7.79, and a survival rate of 8.5 log (cfu/g) for LAB and 8.6 log (cfu/g) for yeasts, which are suitable physicochemical and microbiological characteristics for a functional and probiotic food on the market.

**Author Contributions:** A.Á.C.-I. conceived, designed and performed the experiments; all authors analyzed and evaluated the results, and contributed to paper writing.

**Funding:** The authors thank the support from the Consejo Nacional de Ciencia y Tecnología (CONACyT) in the form (No. 434499) of a scholarship.

**Conflicts of Interest:** The authors declare no conflict of interest.

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
