# Peer review of "Effect of the Freeze-Drying Process on the Physicochemical and Microbiological Properties of Mexican Kefir Grains"

_processes, doi:10.3390/pr7030127_

Reviewer 1 Report

The research studied the optimization of freeze drying parameters with regards to kefir drying. This research is novel to the food industry and provides great insights into Kefir grain shelf life and usage. There are numerous grammatical errors throughout the paper. The introduction is lacking the intention/purpose of this research and needs to be discussed further. The results and discussion were adequate but there are some repetitive lines. A major flaw in the study is regarding the cryoprotectant usage. It is hard to imagine that these microbial cells would survive such extreme temperatures without cryoprotectants? Even if they did, they would be damaged and may not perform as starter cultures.

Line 17-19: What is the purpose of this study?

Line 30: Change it to Kefir grains are starter cultures

Line 33-37: The sentence is too long and confusing. Please break the sentence for clarity.

Line 47-50: Rephrase this. It is grammatically incorrect

The introduction does not discuss the background and the need for this study clearly. Please include more.

Section 2.1: Was there a cryoprotectant used? It is hard to imagine that these cells would survive such extreme temperatures without cryoprotectants? Even if they did, they would be damaged and may not perform as starter cultures.

Line 89: What is inverted plate method?

Line 111: Include the names K1, K2, K3, K4 in Table 1

Line 132: Table 3 shows…

Line 137: most important factors..

Line 179-181: What was the color of powder? What kind of color change would indicate deterioration? Color change is not always corelated with spoilage.

Figure 3: How much of a color difference ∆E of 25 shows in the final product? It would be helpful to include the two pictures to show this difference.

Line 210: bacterial damage

Section 3.4: What about the cryo damage that occurred to the cell? How does that impact the viability to use them as starter cultures? This is a big flaw in the study to not address it.

Line 217: No need to explain R′ every time. Please include this explanation in the beginning of R&D

Line 244: What support materials? Do you mean cryoprotectants?

Line 246: Repetitive

Lines 297-305: What about the future studies?

Author Response

Reviewer 1

We sincerely appreciate your comments about this paper to improve our contribution (425196). The changes in the manuscript are indicated using colored text (blue).

According to reviewer general comment and recommendations, the following modifications have been realized:

·General comment:

This research studied the optimization of freeze drying parameters with regards to kefir drying. This research is novel to the food industry and provides great insights into Kefir grains shelf life and usage. There are numerous grammatical errors throughout the paper. The introduction is lacking, the intention/purpose of this research and needs to be discussed further. The results and discussion were adequate but they are some repetitive lines. A major flaw in the study is regarding the cryoprotectant usage. It is hard to imagine that these microbial cells would survive such extreme temperatures without cryoprotectants? Even if they did, they would be damaged and may not perform as starter culture.

Response: Thank you very much for your comments and suggestions. The comments and suggestions are valuable and very helpful for revising and improving our manuscript. We have revised our manuscript according to your comments and suggestions point by point. 

             Line 17-19: What is the purpose of this study

Response: On lines 17-18, the sentence: “The aim of this research was to investigate how the operating parameters of freeze-drying affected properties of Mexican kefir grains” was rewritten to: The purpose of this study was to investigate how properties of Mexican Kefir grains (MKG) were affected by the operating parameters of freeze-drying.

 Line 30: Change it to Kefir grains are starter cultures

Response: On line 30, it was corrected the text.

Line 33-37: The sentence is too long and confusing. Please break the sentence for clarity.

Response: On lines 33-38 it was rewritten the text.

Line 47-50 Rephrase this. It is grammatically incorrect.

Response: These lines have been carefully revised to improve the grammar and readability.

            The introduction does not discuss the background and the need for this study clearly. Please include more.

Response: On lines 61-67, We have supplemented this information.

Section 2.1: Was there a cryoprotectant used? It is hard to imagine that these cells would survive such extreme temperatures without cryoprotectants. Even if they did, they would be damaged and may not perform as starter cultures.

Response: It is correct; in this study, no cryoprotectant was used during freeze drying. The aim of this study was to evaluate the properties of MKGs without the use of cryoprotectants during freeze-drying. However, under the optimal conditions obtained, the properties of dried MKG were residual moisture 5.03%; aw 0.167; ΔE 7.79; survival of LAB 8.5 Log (CFU/g) and yeasts 8.6 Log (CFU/g) which are physicochemical and microbiological characteristics suitable as a functional and probiotic food for the market. In our future work, we are considering the study of the functionality of the dried MKG as starter culture.

Line 89: What is inverted plate method?

Response: On line 93 “inverted plate method” was corrected to “pour plate Method”

Line 111: include the names K1, K2, K3, K4 in Table 1

Response: the names K1, K2, K3, K4 were included in Table 1.

Line 132: Table 3 shows….

Response: On line 136, “ The Table 3 shows” was corrected to “Table 3 shows”

Line 137: most important factors

Response: on line 141, it was added “s” to factor

Line 179-181: what was the color of powder? What kind of color change would indicate deterioration? Color change is not always correlated with spoilage. Figure 3: How much of a color difference ΔE of 25 shows in the final product? If would be helpful to include the two pictures to show this difference.

Response: Thank you very much for your comment and suggestion. On lines 202-205, three pictures to show visual color of powder and color difference among some experiments were included in the revised manuscript (Figure 4). We are not very sure if you mean the Table or Figure 3, maximum ΔE in the final product was 22.17 (Table 3, Experiment 11) and its picture was included in the Figure 4. Preferred colors of a dried product are those closest to the original color of fresh sample.

We agree with color change is not always correlated with spoilage, however, color of any dried product is an important factor to consider, as it is one of the main attributes to evaluate quality and is the first quality judgement made by a consumer on a food at the point of sale.

Line 210 bacterial damage

Response: on line 208, “bacterial damager” was corrected to “bacterial damage”

Section 3.4 what about the cryo damage that occurred to the cell? How does that impact the viability to use them as started cultures? This is a big flaw in the study to not address it.

Response: It is correct, in this study did not evaluate the cryo damage on cell. In future work this is already being considered.

Line 217: no need to explain R’ every time. Please include this explanation in the beginning of R&D.

Response: We have adjusted the text as suggested.

Line 244: What support materials? Do you mean cryoprotectants?

Response: We want to say cryoprotectants. ”support materials” was corrected to “cryoprotectants”.

Line 246: Repetitive

Response: it was eliminated repetitive text as suggested.

Line 297-305: What about the future studies?

Response: we hope to continue with future stability studies during storage, evaluating different storage conditions. We think this is not necessary to include in conclusions

Reviewer 2 Report

The manuscript must be revised by a native english speaker. Due to inappropriate english language some sentences do not make sense. Please revise carefully references. In annex I make some suggestions to improve the manuscript.

Author Response

Reviewer 2

We sincerely appreciate your comments about this paper to improve our contribution (425196).

The changes in the manuscript are indicated using colored text (yellow).

According to reviewer comments, the following modifications have been realized:

1.      We have corrected the text as suggested. We agree with you that some statements were lightly ambiguous than intended and we have adjusted the text to be clearer. We have improved our English writing in lines 49-55, 149-152 (before 145-148), 179-183 (before 183-186), 185-187 (before 188-191), 205-207, 220-222 (before 225-228), 240-241 (before 243-244), 254-256 (before 259-261), 272-273, 283-284. In addition, our manuscript has been reviewed by a native speaker to improve readability.

2.      We agree with you in line 217 (before 222), given that freezing situations are not comparable with our results because at - 196 ºC, freezing was done by immersion in liquid nitrogen. Therefore, this comparation has been deleted from this version of the manuscript.

3.      For the suggestion about underline w in aw, we are afraid that we cannot underline text because acronym of water activity is showed for aw in literature.

5.      In lines 208 and 277 (before 213, 283) the valor 7 Log CFU/g was corrected to 6 Log CFU/g.

Sincerely,

G. Luna-Solano

Round  2

Reviewer 1 Report

All the technical comments were addressed.

The paper still has several grammatical errors. It must be edited by professional editors before it can be published. 

Author Response

Dear editor

The paper was already edited by professional editors of MDPI, we are sending it the final version

Regards,

Dr. G. Luna-Solano
